# How mono- and diphosphine ligands alter regioselectivity of the Rh-catalyzed annulative cleavage of bicyclo[1.1.0]butanes

Pan-Pan Chen [1], Peter Wipf [2]✉ & K. N. Houk [1]✉

Rh(I)-catalyzed cycloisomerizations of bicyclo[1.1.0]butanes provide a fruitful approach to cyclopropane-fused heterocycles. Products and stereochemical outcome are highly dependent on catalyst. The triphenylphosphine (PPh$_3$) ligand provides pyrrolidines, placing substituents *anti* to the cyclopropyl group. The 1,2-bis(diphenylphosphino)ethane (dppe) ligand yields azepanes with substituents *syn* to the cyclopropyl group. In this work, quantum mechanical DFT calculations pinpoint a reversal of regio- and diastereoselectivity, suggesting a concerted (double) C−C bond cleavage and rhodium carbenoid formation, driven by strain-release. The ligand-influenced cleavage step determines the regioselectivity of carbometalation and product formation, and suggests new applications of bicyclobutanes.

Strain-release driven transformations are valuable methods for the synthesis of polycyclic fused and bridged ring systems[1,2]. Bicyclo[1.1.0]butanes (BCBs) are the most highly strained of fused bicyclic ring systems and have unique chemical and electronic properties[1]. Along with [1.1.1]propellanes, 1-azabicyclo[1.1.0]butanes and bicyclo[2.1.0]pentanes, BCBs are "spring-loaded" molecules[3,4]. Leveraging the >60 kcal/mol strain energy and "olefinic" properties of the central C−C σ bond[5–7], BCBs display distinct behavior towards nucleophiles, electrophiles, and radicals[2,8–13]. Because of their reactivity and potential use as bioisosteres in medicinal chemistry, they are attracting considerable attention[14–16]. In particular, transition metal-catalyzed transformations of BCBs are of great interest[17–29]. In 2019, the Aggarwal group reported a synthesis of difunctionalized cyclobutyl boronates by virtue of a carbopalladation of the central C−C σ-bond of BCB boronate complexes, facilitated by a 1,2-metalate rearrangement[30]. Later, Gryko and coworkers presented a polarity-reversal strategy for the functionalization of electrophilic BCB molecules via light-driven cobalt catalysis. Nucleophilic cyclobutyl radicals and the Co(II) catalyst are generated upon homolytic cleavage of Co(III)−alkyl species by irradiation. The resulting cyclobutyl radicals are then trapped by different electrophiles[31]. By employing Cp*Rh(III) as catalyst and taking advantage of the strain-release from BCB derivatives, the Glorius group recently developed a highly

diastereoselective and E-selective three-component method to construct quaternary carbon centers[32].

Catalyst-directed divergent cycloisomerizations of BCBs are promising methods to generate structurally complex 5–7-membered heterocyclic products from readily available starting materials simply by modifying the catalyst system. In 2008, the Wipf group showed that phosphine ligands have a decisive effect on the regioselectvity of Rh(I)-catalyzed cycloisomerizations of bicyclobutanes (Fig. 1)[33,34]. In the presence of catalytic [Rh(C$_2$H$_4$)$_2$Cl]$_2$/PPh$_3$, the reaction generates five-membered cyclopropyl-fused pyrrolidines, placing the C(2)-substituent and the cyclopropyl ring on opposite sides of the 5-membered ring in 2. In contrast, the catalytic [Rh(CO)$_2$Cl]$_2$/dppe (dppe, 1,2-bis(diphenylphosphino)ethane) system delivers a high regio- and diastereoselectivity for azepane formation with a *syn*-orientation of the C(2)-substituent and the cyclopropyl group on the 7-membered ring in 3.

Although other rearrangements of BCBs have been catalyzed by metals[17–25], the mechanistic details of these transformations are still unknown. For the Rh(I)-catalyzed cycloisomerizations of BCBs (Fig. 1), the ligand effects on regioselectivity were not rationalized based on known mechanistic models[30–32,34]. Furthermore, the origins of the observed diastereoselectivity are also intriguing. To unravel the intricacies of ligand-dependent divergent reactions and the origins of

[1]Department of Chemistry and Biochemistry, University of California, Los Angeles, CA 90095, USA. [2]Department of Chemistry, University of Pittsburgh, 219 Parkman Avenue, Pittsburgh, PA 15260, USA. ✉e-mail: pwipf@pitt.edu; houk@chem.ucla.edu

stereoselectivity, we undertook a computational study of this transformation.

Here, we report the reaction mechanism and origins of ligand-dependent regio- and diastereoselectivity of Rh(I)-catalyzed cycloisomerizations of bicyclo[1.1.0]butanes. A concerted (double) C–C bond cleavage and Rh-carbenoid formation process, which determines the regioselectivity and the reaction paths, is discovered and investigated. The newly revealed mechanistic underpinnings of selectivity in Rh(I)-catalyzed cycloisomerizations will provide guidance for the future rational design of new regio- and stereoselective reactions involving transition metal-catalyzed conversions of "spring-loaded" molecules.

## Results
### Computational study of Rh(I)/PPh₃-catalyzed cycloisomerizations

Neutral (Fig. 2) and ionic (Fig. 3) mechanisms were envisioned. Starting from L_nRh(I) species I in Fig. 2, substrate (II) coordination occurs to generate the π-coordinated complex, III. *Endo* oxidative addition then occurs to cleave the central C–C bond in the BCB, generating Rh(III) species IV. We hypothesized that IV undergoes a subsequent rearrangement to carbenoids V or IX as a function of added ligand (path A

vs. path D). A related tricyclic intermediate IV has been characterized for reactions of bicyclo[1.1.0]butanes with Pt(II) complexes. Experimental observations also indicate that IV may be in equilibrium with its isomers formed by insertion of the metal into the lateral bonds of BCB

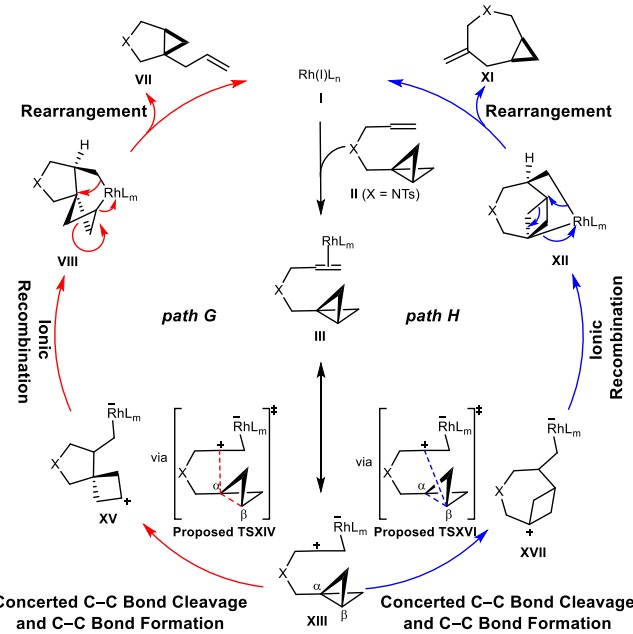

**Fig. 3 | Ionic mechanisms for Rh(I)/PPh₃-catalyzed cycloisomerization of bicyclo[1.1.0]butane.** Starting from the active catalyst I, two different catalytic cycles (path G and path H) were proposed to form five- (VII) or seven-membered (XI) product. Path G leads to the formation of VII, and path H gives rise to the generation of XI. Path G, solid red curve; Path H: solid blue curve.

**Fig. 1 | Ligand-controlled reaction outcome.** Regio- and diastereoselectivity of Rh(I)-catalyzed cycloisomerizations of bicyclo[1.1.0]butanes. Under the catalysis of Rh(I)/PPh₃, the reaction mainly generates five-membered product (2). In contrast, employing Rh(I)/dppe as catalyst favors seven-membered product (3) formation.

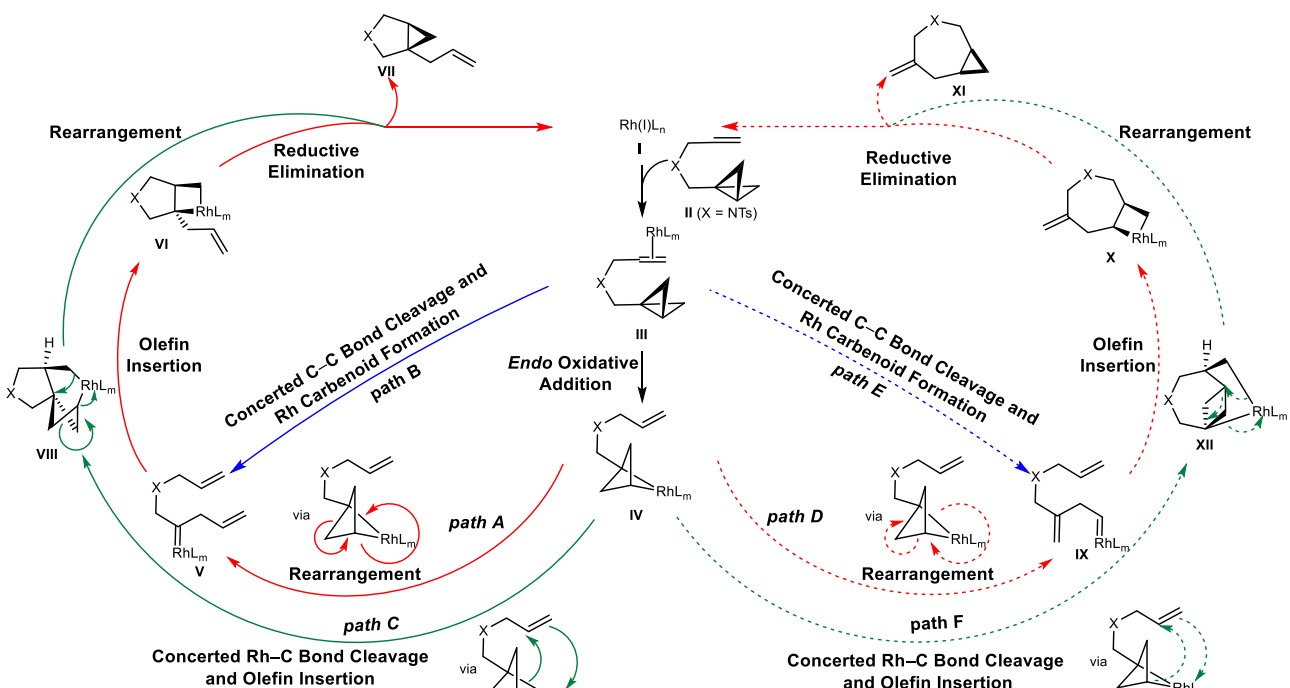

**Fig. 2 | Neutral mechanisms for Rh(I)/PPh₃-catalyzed cycloisomerization of bicyclo[1.1.0]butane.** Starting from the active catalyst I, six different catalytic cycles (path A to path F) were proposed to form five- (VII) or seven-membered (XI) product. Paths A, B, and C lead to the formation of VII, and paths D, E and F give rise

to the generation of XI. Path A, solid red curve; Path B, solid blue curve; Path C, solid green curve; Path D, dotted red curve; Path E, dotted blue curve; Path F, dotted green curve.

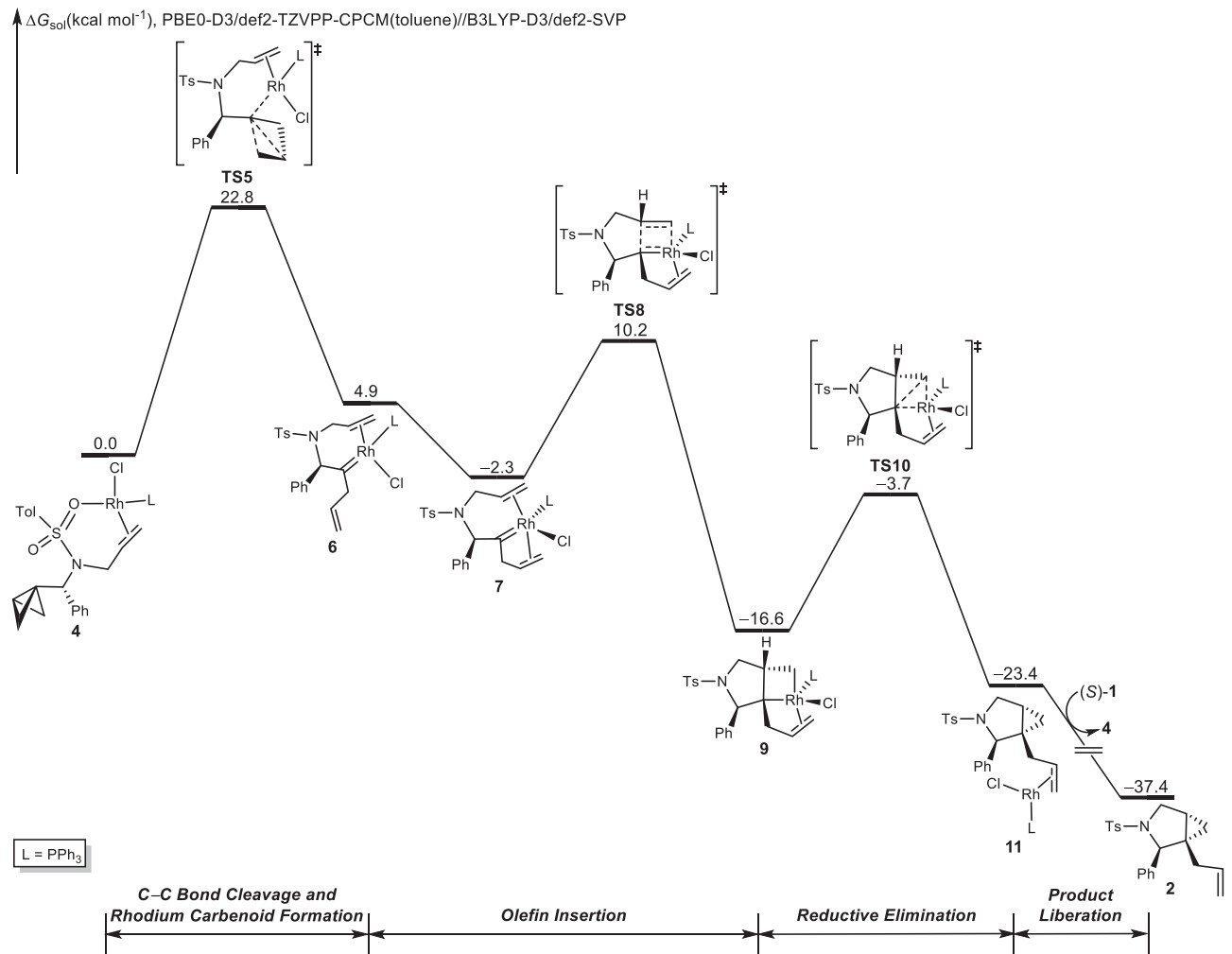

**Fig. 4 | Computational investigation of Rh/PPh₃ case.** DFT-computed free-energy changes of Rh(I)/PPh₃-catalyzed cycloisomerization of bicyclo[1.1.0]butane. This energy profile corresponds to path B shown in Fig. 2. Calculations were performed at the PBE0-D3/def2-TZVPP-CPCM(toluene)//B3LYP-D3/def2-SVP level of theory. Free energies ($\Delta G_{sol}$) are in kcal/mol.

**II**[35,36]. Alternatively, the $L_nRh(I)$ species **I** can react with BCB **II** through a concerted process in which the central C−C bond cleavage and rhodium carbenoid formation occur at the same time to form carbenoids **V** or **IX** directly depending on the choice of ligand (path B vs path E). Paths A and B converge on internal carbenoid (**V**) formation, and paths D and E converge on external carbenoid (**IX**) formation. Olefin insertion into the carbenoid occurs to give cyclometallation intermediate **VI** or **X**. The final step in the catalytic cycle would be the release of the metal complex through C−C reductive elimination (intramolecular cyclopropanation), furnishing pyrrolidine **VII** or azepane **XI**. Moreover, starting from intermediate **IV**, a concerted Rh−C bond cleavage and olefin insertion can occur simultaneously through path C or path F, generating intermediate **VIII** or **XII** (paths C and F bypass the generation of rhodium carbenoids). Subsquent rearrangement occurs to deliver five- (**VII**) or seven-membered (**XI**) product.

Possible catalytic cycles involving zwitterionic intermediates are shown in Fig. 3. Starting from the active catalyst $L_nRh(I)$ species **I**, substrate (**II**) coordination occurs to generate complex **III**. This complex can be envisioned as a zwitterionic isomer **XIII**, since **XIII** is likely to be a major resonance contributor to **III**. From **XIII**, cleavage of the central C−C bond in the BCB along with C−C(α) (via a proposed **TSXIV**) or C−C(β) bond formation (via a proposed **TSXVI**) would generate zwitterions **XV** or **XVII**, from which ionic recombination occurs to produce **VIII** or **XII**. Subsequent rearrangements form pyrrolidine (**VII**) or azepane (**XI**), as well as the active catalyst (**I**), completing the catalytic cycle.

To study the proposed mechanisms and the origins of regio- and diastereoselectivity in this transformation, we performed computations using the exact substrate (($S$)-**1**) and ligands (PPh₃ and dppe) as the experimental study (Fig. 1)[33]. To discriminate between the mechanistic possibilities, we began by evaluating the possible C−C bond cleavage pathways that could initate the catalytic cycles. QM calculations (computational details are included in the Supplementary Information) were performed using the B3LYP-D3/def2-SVP method for geometry optimization and the PBE0-D3/def2-TZVPP-CPCM(toluene) method for single-point energy calculation.

The free-energy changes of the most favorable pathway of the Rh(I)/PPh₃-catalyzed cycloisomerization of BCB **1** are shown in Fig. 4. Density functional theory (DFT)-optimized structures of selected intermediates and transition states are shown in Supplementary Fig. 1. Based on the experimental reaction conditions[33], the reference point of the catalyst was found (Supplementary Fig. 2) to be **4**, the catalyst resting state. Cleavage of the central C−C bond of BCB in **4** involves a concerted transition state (**TS5**), in which C−C bond cleavage and Rh-carbenoid formation occur simultaneously to produce the internal carbenoid **6**. We also located similar C−C bond cleavage transition states without alkene assistance or with oxygen (from the tosyl group) coordination, but these transition states were all higher in energy than **TS5** (Supplementary Fig. 3). Completion of the catalytic cycle requires, in broad strokes, olefin insertion and reductive elimination. On the basis of our calculations, this process proceeds first by coordination of

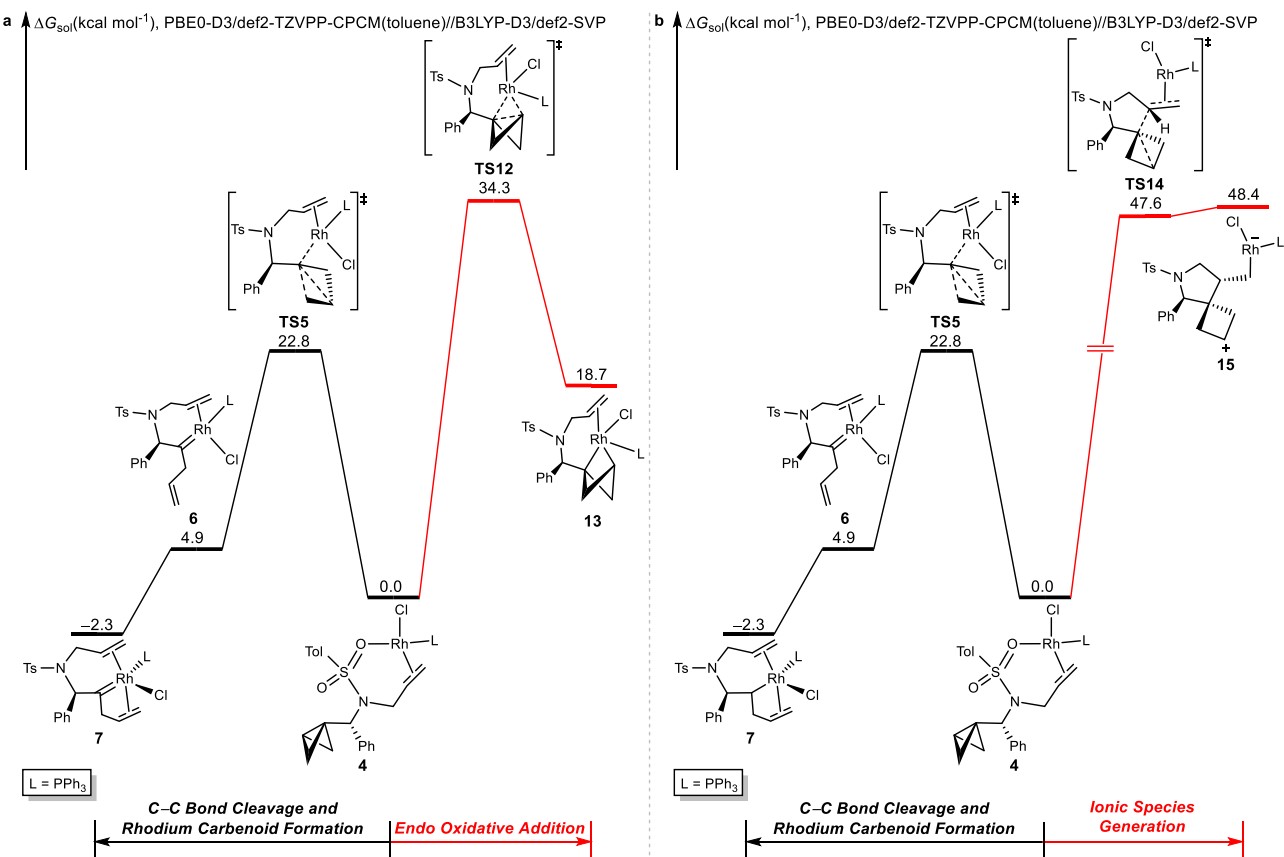

**Fig. 5 | DFT analysis of the alternative pathways. a** Free-energy changes of the competing rhodium carbenoid fromation (path B in Fig. 2, black line) and *endo* oxidative addition (transformations from **III** to **IV** in Fig. 2, red line) from intermediate **4**. **b** Free-energy changes of the competing rhodium carbenoid fromation

(path B in Fig. 2, black line) and ionic species generation (path G in Fig. 3, red line) from intermediate **4**. Calculations were performed at the PBE0-D3/def2-TZVPP-CPCM(toluene)//B3LYP-D3/def2-SVP level of theory. Free energies ($\Delta G_{sol}$) are in kcal/mol.

uncomplexed olefin in **6** to the metal center, forming a stable intermediate **7**. This complex (**7**) undergoes a facile olefin insertion via **TS8** to generate the cyclometallation intermediate **9**. The alternate olefin insertion transition state without alkene coordination is less favorable compared to **TS8** (Supplementary Fig. 4). Subsequent reductive *syn*-elimination via **TS10** generates the new C−C bond and leads to the product-coordinated complex **11**. Similar to olefin insertion via **TS8**, alkene coordination promotes the reductive elimination, leading to C−C bond formation (Supplementary Fig. 5). Product release from complex **11** generates the five-membered ring **2** and active Rh(I) catalyst (**4**) for the next catalytic cycle. Intrinsic reaction coordinate (IRC) analyses of key transition states were performed to verify their positions on the free-energy surface (Supplementary Fig. 6).

Alternative catalytic cycles involve initial central C−C bond *endo* oxidative addition (Fig. 5a), and the free-energy changes of the competing pathways from the substrate-ligated catalyst, intermediate **4**, are shown in Fig. 5. The proposed propellane-like *endo* oxidative addition transition state (**TS12**) to break the central C−C bond is 11.5 kcal/mol less favorable than an irreversible lateral C−C bond cleavage via **TS5**. The *endo* oxidative addition of the C−C bond is not operative in that it will produce a large ring tension, as can be seen from the energy of rhodium-propellane **13**, which is 18.7 kcal/mol higher in energy than the reference point **4**. We also considered the possibility of a coordination of the oxygen atom of the tosyl group to the rhodium promoting the *endo* oxidative addition, as well as the same process without the assistance of alkene; however, these hypothetical processes are also not feasible (Supplementary Fig. 7). The unique ring structure (ring strain) of the BCB determines the bond activation mode under Rh(I) catalysis.

Subsequently, we investigated the possibility of an ionic mechanism leading to an opening of the central C−C bond of BCBs under rhodium catalysis. However, this proposed pathway proved not to be feasible. The free-energy changes of two competing pathways from intermediate **4** are shown in Fig. 5b. From **4**, C−C bond cleavage via **TS5** is facile with a barrier of 22.8 kcal/mol, while C−C bond cleavage through **TS14** is significantly less favorable, with a reaction energy barrier of up to 47.6 kcal/mol. The alternative transition states to generate ionic species are even less favorable compared to **TS14** (Supplementary Fig. 8). Both **TS14** and **15** are highly disfavored because the high-polarity zwitterionic species is likely disfavored in the experimentally employed low-polarity toluene solvent. Thus, sequential transformations involving charge-separated intermediates are unlikely to be operative for this cycloisomerization reaction.

The analyses shown in Fig. 5 suggested that under Rh/PPh₃ catalysis, the cycloisomerization reaction started with concerted C−C bond cleavage and rhodium carbenoid formation via **TS5**. This process is favored because it bypasses the formation of rhodium intermediates with high ring strain and avoids the formation of polar zwitterionic species. In turn, the strain-release from the BCB, along with the formation of a neutral carbenoid, represents the driving force of this concerted process through **TS5**.

The mode of bond activation occurring in **TS5** has the potential to contribute to the enrichment of two research topics that have long been contemplated in synthetic chemistry: (1) transition metal-catalyzed carbon–carbon bond activation[37–40] in which the cleavage of C−C bonds with catalysts is achieved through either oxidative addition[41–44] or β-carbon elimination[45–47]. The concerted double C−C bond activation (via **TS5**) reported in this work can be considered as an

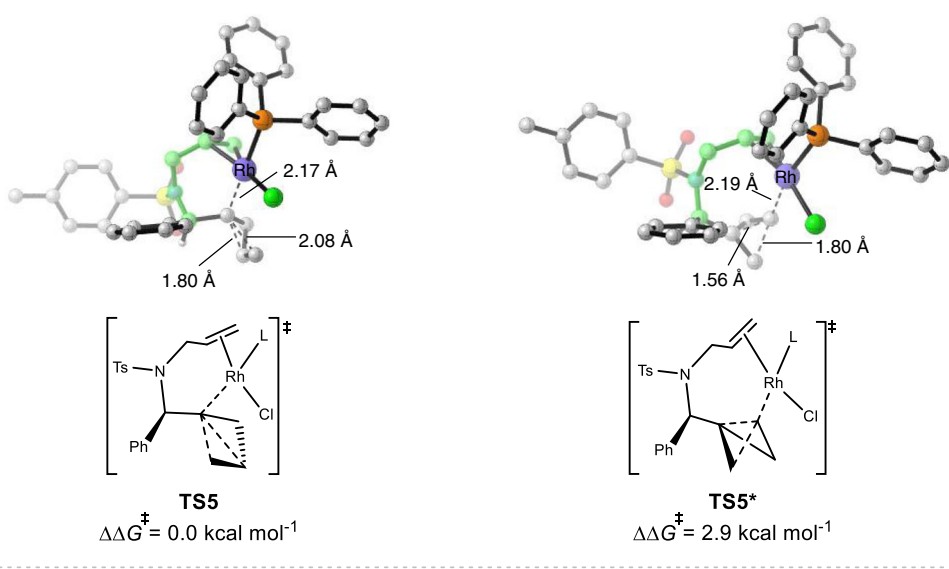

**a** Competitive rhodium carbenoid formation transition states

**TS5**
$\Delta\Delta G^{\ddagger}$ = 0.0 kcal mol$^{-1}$

**TS5\***
$\Delta\Delta G^{\ddagger}$ = 2.9 kcal mol$^{-1}$

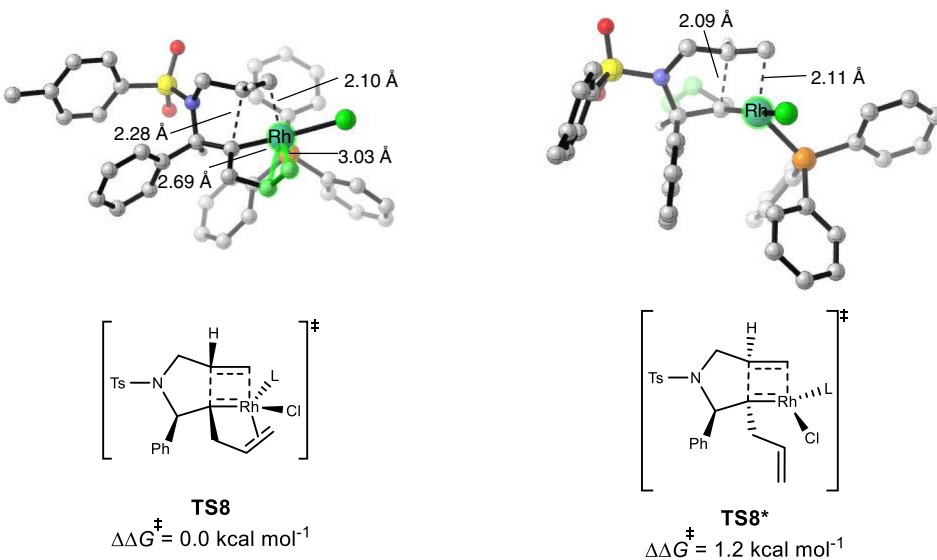

**b** Competitive olefin insertion transition states

**TS8**
$\Delta\Delta G^{\ddagger}$ = 0.0 kcal mol$^{-1}$

**TS8\***
$\Delta\Delta G^{\ddagger}$ = 1.2 kcal mol$^{-1}$

**Fig. 6 | DFT analysis of the origins of regio- and diastereoselectivity for Rh(I)/PPh₃-catalyzed cycloisomerizations.** DFT-optimized structures of **a** rhodium carbenoid formation and **b** olefin insertion transition states involved in Rh/PPh₃ case. L = PPh₃. Calculations were performed at the PBE0-D3/def2-TZVPP-CPCM(toluene)//B3LYP-D3/def2-SVP level of theory.

alternative approach to catalytic C−C bond cleavage. (2) The formation of a carbenoid complex from highly reactive compounds, such as diazoalkanes[48–50], containing weak C−X bonds that are easily cleaved by the metal catalyst[51]. Our study demonstrates that the carbenoid formation can also be accomplished from reactive hydrocarbons. We envision that this fundamental bond activation pattern (through **TS5**) may be generally relevant to other transition metals and "spring-loaded" molecules.

On the basis of the calculated free-energy changes of the entire catalytic cycle (Fig. 4), the on-cycle resting state is the substrate-coordinated Rh(I) intermediate **4**, and the rate- and regioselectivity-determining step is the concerted C−C bond cleavage and rhodium carbenoid formation via **TS5** with an overall barrier of 22.8 kcal/mol. The diastereoselectivity-determining step is the olefin insertion via **TS8**. Subsequently, a reductive elimination step is both facile and does not alter the configuration at the stereogenic carbon, producing the

five-membered ring product and placing cyclopropyl ring and the phenyl group at the C(2) position into an *anti*-relationship.

## Origins of regioselectivity of Rh(I)/PPh₃-catalyzed cycloisomerizations

The free-energy profile of Rh(I)/PPh₃-catalyzed cycloisomerization of BCB **1** (Fig. 4) indicated that the concerted C−C bond cleavage and rhodium carbenoid formation is irreversible and determines the overall regioselectivity of the transformation. Figure 6a shows the competing transition states of C−C bond activation that determine the regioselectivity. The C−C bond cleavage, along with carbenoid formation, can occur at the internal carbon through transition state **TS5**, or at the external carbon via transition state **TS5\***. **TS5\*** is less favorable as compared to **TS5** (2.9 kcal/mol in terms of Gibbs free energy, Fig. 6a), indicating that the C−C bond cleavage occurs with preferred internal carbenoid formation.

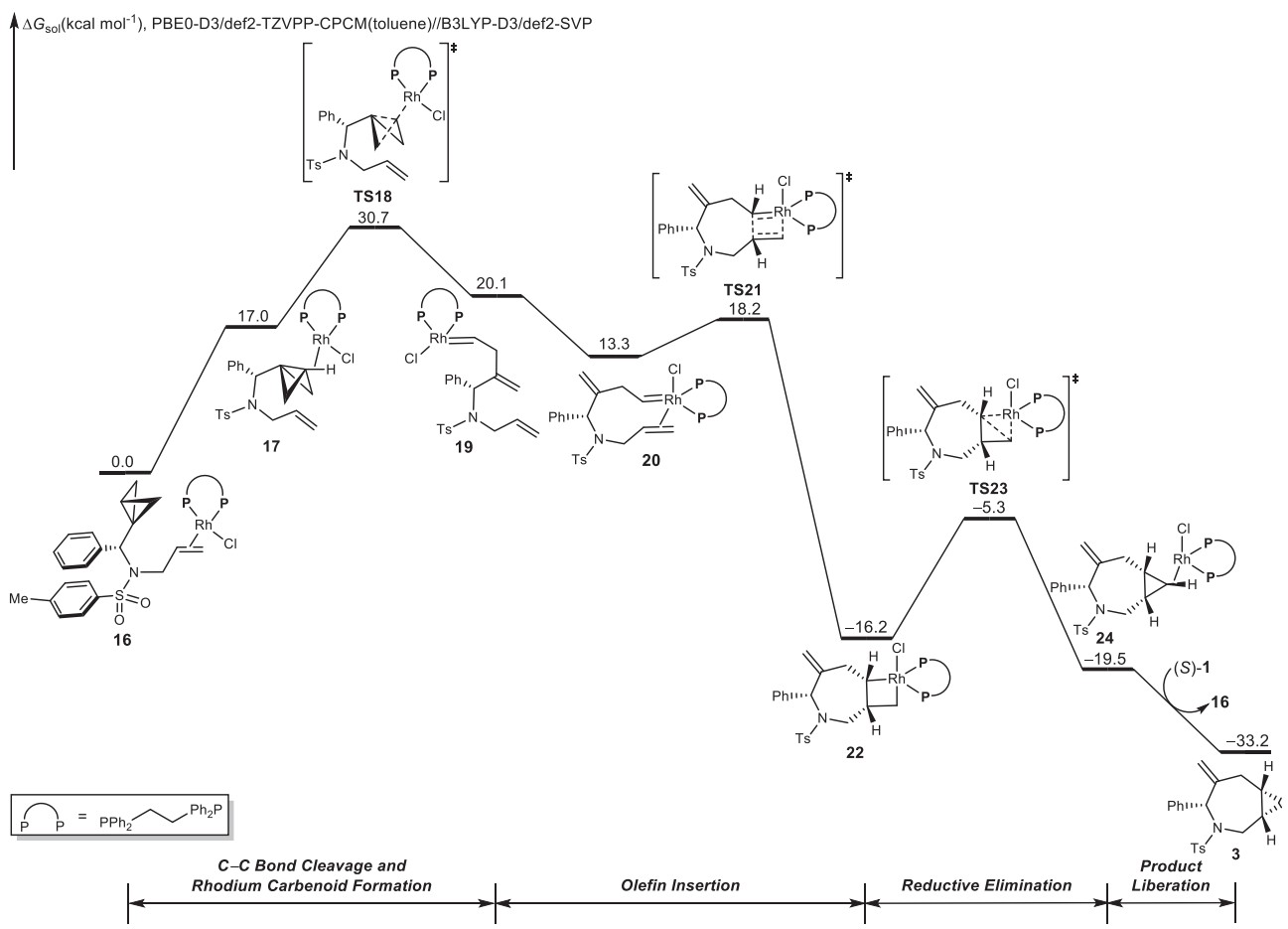

**Fig. 7 | Computational investigation of Rh/dppe case.** DFT-computed free-energy changes of Rh(I)/dppe-catalyzed cycloisomerization of bicyclo[1.1.0] butane. This energy profile corresponds to path E shown in Fig. 2. Calculations were performed at the PBE0-D3/def2-TZVPP-CPCM(toluene)//B3LYP-D3/def2-SVP level of theory. Free energies ($\Delta G_{sol}$) are in kcal/mol.

The Rh(I)/PPh$_3$-catalyzed cycloisomerizations of BCBs are intramolecular reactions, in which the concerted C−C bond cleavage and rhodium carbenoid formation occur with the assistance of the alkene moiety in the substrate. Based on this scenario, we assumed that the linker (the green-highlighted part in **TS5** and **TS5***, Fig. 6a) has a significant effect on regioselectivity: under the influence of the linker, the Rh catalyst tends to attack the adjacent internal carbon via **TS5**, whereas in **TS5***, the Rh catalyst, in order to attack at the external carbon, restricts the conformation of the linker. To support this hyphothesis, we calculated the energy difference of the molecular skeleton highlighted in green in **TS5** and **TS5*** (Fig. 6a), by replacing the PPh$_3$RhCl(BCB) moiety with a hydrogen atom. Indeed the linker in **TS5** is 2.3 kcal/mol more stable than that in **TS5***, due primarily to the distortions of dihedral angles in the linker of **TS5***. Details are included in the Supplementary Fig. 9.

## Origins of diastereoselectivity of Rh(I)/PPh$_3$-catalyzed cycloisomerizations
On the basis of the free-energy profile of the Rh(I)/PPh$_3$-catalyzed cycloisomerization of BCB **1** (Fig. 4), the olefin insertion through **TS8** is irreversible and determines the overall diastereoselectivity of the cycloisomerization. Figure 6b shows the possible competing transition states that could result in the formation of different diastereoisomers. The olefin insertion can occur via **TS8**, in which the phenyl and allyl groups are positioned on the same face of the pyrrolidine. Alternatively, the olefin insertion can occur through **TS8*** with phenyl and allyl groups on opposite sides. **TS8** is 1.2 kcal/mol more favorable than

**TS8*** in terms of Gibbs free energy, indicating that olefin insertion tends to occur through **TS8**, resulting in the formation of the experimentally observed *syn*-product. **TS8** is lower in energy than **TS8*** because of the stable rhodium-alkene coordination (highlighted in green), which is present in the former but absent in the latter, thus providing additional stabilization energy in **TS8** (Supplementary Fig. 10).

## Computational study of Rh(I)/dppe-catalyzed cycloisomerizations
Based on the mechanistic scenario obtained from the Rh(I)/PPh$_3$ case, we next investigated the Rh(I)-catalyzed cycloisomerizations of BCBs using the dppe ligand. The Gibbs free-energy changes of the most favorable pathway that produces the azepane are shown in Fig. 7, and the optimized structures of selected intermediates and transition states are illustrated in Supplementary Fig. 11. Starting from substrate-coordinated π-complex **16**, a direct attack from the rhodium catalyst to the external carbon of the BCB occurs through **TS18** to produce an external Rh-carbenoid species **19**. Alternative transition states to cleave the central C−C bond of BCB are less favorable (Supplementary Fig. 12). The activation mode through **TS5** or **TS5*** involves the alkenyl group acting as auxiliary ligand bridging to the metal center. By contrast, the bidentate ligand dppe coordinatively saturates the Rh, and thus does not require the assistance of the alkene during the cleavage of the carbon−carbon bond. Similar to the Rh/PPh$_3$ case, the concerted C−C bond cleavage and Rh-carbenoid formation via **TS18** under the catalysis of Rh/dppe is also irreversible, suggesting that the regioselectivity

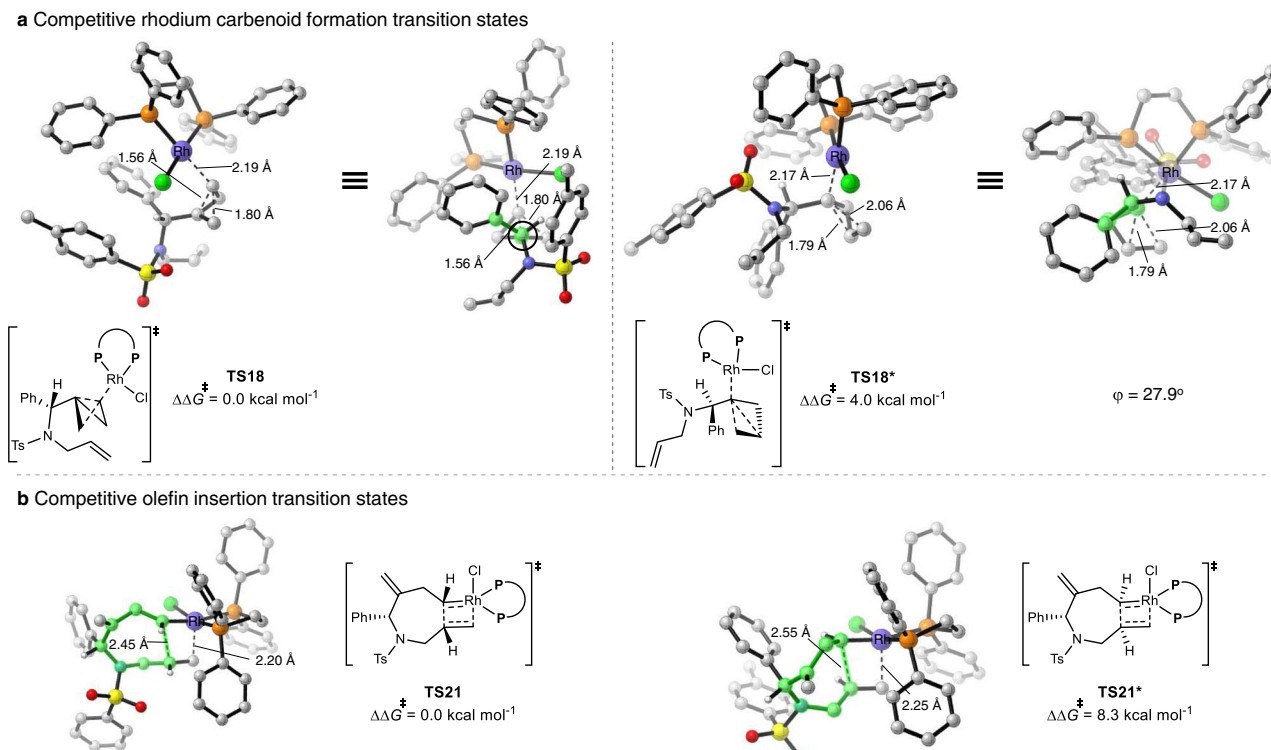

**Fig. 8 | DFT analysis of the origins of regio- and diastereoselectivity for Rh(I)/dppe-catalyzed cycloisomerizations.** DFT-optimized structures of **a** rhodium carbenoid formation and **b** olefin insertion transition states involved in Rh/dppe case. Calculations were performed at the PBE0-D3/def2-TZVPP-CPCM(toluene)//B3LYP-D3/def2-SVP level of theory. φ, dihedral angle.

of this step is kinetically determined. From **19**, a conformational change produces a stable intermediate **20**, from which olefin insertion occurs through **TS21** to generate the metallacycle intermediate **22**. Subsequently, C−C bond formation and reductive elimination via **TS23** produces the product-coordinated complex **24**. The final ligand exchange can release the seven-membered ring product **3** and regenerate the active rhodium(I) catalyst (**16**) for the next catalytic cycle. IRC analyses of key transition states were performed to verify their positions in the free-energy surface (Supplementary Fig. 13).

For the catalytic cycle with the dppe ligand, the on-cycle resting state is intermediate **16**, and the rate- and regioselectivity-determining step is the concerted C−C bond cleavage and Rh-carbenoid formation via **TS18** with a 30.7 kcal/mol overall barrier (computed energy profiles (Fig. 4 and Fig. 7) indicate that the reaction barriers for the corresponding rate-determining steps are 22.8 and 30.7 kcal/mol for Rh(I)/PPh₃ and Rh(I)/dppe cases, respectively. This energy difference implies that Rh(I)/dppe is less active than Rh(I)/PPh₃ for the cycloisomerization of BCB, which is consistent with the experimental observations[33]. The stereoselectivty-determining step is the alkene carbometalation via **TS21**, and the exergonicity of the overall transformation (33.2 kcal/mol) does not affect the catalytic turnover, since no catalyst-poisoning species are involved.

### Origins of regioselectivity of Rh(I)/dppe-catalyzed cycloisomerizations

Similar to the Rh/PPh₃ case, the concerted C−C bond cleavage and Rh-carbenoid formation via **TS18** is irreversible and determines the overall regioselectivity of the reaction (Fig. 7). Figure 8a shows the competing transition states for the Rh(I)/dppe-catalyzed central C−C bond activation of BCBs. The C−C bond cleavage, along with the Rh-carbenoid formation, can occur at the internal carbon through transition state **TS18***, or at the external carbon via transition state **TS18**. **TS18** is 4.0 kcal/mol more favorable than **TS18*** in terms of Gibbs free energy,

indicating that the C−C bond cleavage occurs with exclusive external carbenoid formation.

Rh(I)/dppe-catalyzed cycloisomerizations of BCBs are intermolecular reactions, and the Rh catalyst tends to attack the external carbon via **TS18**. This is facilitated by the fact that the substituents on the highlighted carbon atom (black circle) in **TS18** can easily assume a staggered conformation with respect to the substituents of the carbon atom in the back of the Newman projection. In contrast, in the competing transition state **TS18***, the phenyl group in the substrate almost eclipses the lateral C−C bond of the BCBs (the highlighted dihedral angle is 27.9°, Fig. 8a). Furthermore, if a smaller substituent such as hydrogen is used to replace the phenyl group in the substrate to construct **TS18-model** and **TS18*-model**, the energies of **TS18-model** and **TS18*-model** are similar, consistent with our rationalization (Supplementary Fig. 14).

### Origins of diastereoselectivity of Rh(I)/dppe-catalyzed cycloisomerizations

On the basis of the free-energy profile of the Rh(I)/dppe-catalyzed cycloisomerization of BCB **1** (Fig. 7), alkene insertion through **TS21** is irreversible and determines the overall diastereoselectivity of the cycloisomerization. Figure 8b shows the possible competing transition states that could result in the formation of different diastereoisomers. The alkene carbometalation can occur via **TS21**, in which the phenyl ring and the emerging cyclopropyl group are on the same face of the azepane. Alternatively, the C−C bond formation can occur through **TS21*** where the phenyl ring and the emerging cyclopropyl group are on opposite sides. **TS21** is 8.3 kcal/mol more stable than **TS21***, indicating that alkene carbometalation exclusively proceeds through **TS21**, forming the experimentally observed product (Fig. 8b).

Fragment analyses (details are included in the Supplementary Fig. 15) indicated that the energy difference of the competing transition states mainly results from the ring strain of the seven-membered

ring (highlighted in green, Fig. 8b), which is larger in **TS21***. The overall selectivity is thus readily rationalized since, in **TS21**, the seven-membered ring can assume a pseudo-chair conformation, while in **TS21*** the seven-membered ring needs to form a pseudo-boat conformation to accommodate the bond formation process[52].

## Discussion

Reaction mechanism and origins of ligand-dependent regio- and diastereoselectivity of Rh(I)-catalyzed cycloisomerizations of bicyclo[1.1.0]butanes have been elucidated with DFT computations. With Rh/PPh$_3$ and Rh/dppe, concerted C−C bond cleavage and Rh-carbenoid formation determine the regioselectivity. In the Rh/PPh$_3$ case, the key difference between the two competing transition states is the skeletal deformation. The internal carbenoid formation transition state is favored due to lower strain. In contrast, with Rh/dppe, the external carbenoid formation transition state is favored since the substituents can assume a staggered conformation with respect to the breaking of the central C−C bond of the bicyclobutane. The diastereoselectivity arises from a stabilizing alkene coordination and a pseudo-chair seven-membered ring conformation in the favored transition states for the Rh/PPh$_3$ and Rh/dppe cases, respectively.

In the regioselectivity-determining C−C bond cleavage and Rh-carbenoid formation step with the PPh$_3$ ligand, the alkene auxiliary ligand coordinates with the metal, facilitating carbon–carbon bond cleavage, and guiding carbenoid formation. The dppe ligand saturates the metal coordination site, and the alkene in the substrate is not necessary as assisting ligand. Accordingly, different factors control the regioselectivity, and the reaction paths. In the case of Rh/PPh$_3$, carbenoid formation is an intramolecular reaction, and internal carbenoid formation is favored; thus, for monodentate ligands, electronic effects have a greater influence on selectivity than steric effects. In contrast, for the Rh/dppe case, carbenoid formation is an intermolecular reaction. Consequently, the catalyst tends to attack the external carbon, minimizing steric hindrance. Therefore, for bidentate ligands, steric effects contribute more to selectivity than electronic effects.

These computational results reshape our understanding of the reactions of BCB derivatives with rhodium catalysts and provide a rationalization of the critical role of the phosphine ligands. We elucidate the details of the double C−C bond cleavage coupled with rhodium carbenoid formation, a pathway that is uniquely feasible in such "spring-loaded" molecules. Insights into the transition states that lead to regio- and stereoselective product formations, and their dependence on the nature of the ligands, will facilitate the design of new catalytic reactions of highly strained carbocyclic building blocks, including the consideration of internally coordinating auxiliary ligands that might influence new reaction pathways to rearrangement products.

## Methods

The detailed computational methods were provided in Supplementary Information.

## Data availability

All data supporting the findings of this study are available within the paper and its Supplementary Information or from the corresponding authors upon request. Supplementary Data 1 contains the cartesian coordinates of calculated structures.

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

## Acknowledgements

We are grateful to the National Science Foundation (CHE-1764328 to K.N.H. and CHE-0910560 to P.W.) for financial support of this research. Calculations were performed on the IDRE Hoffman2 cluster at the University of California, Los Angeles, and the Extreme Science and Engineering Discovery Environment (XSEDE), which is supported by the National Science Foundation (OCI–1053575).

## Author contributions

P.W. and K.N.H. conceived the project. P.C. performed the DFT calculations. P.C., P.W., and K.N.H. supervised the research and co-wrote the manuscript.

## Competing interests

The authors declare no competing interests.
