## [Peer Review File · Nature Communications]

REVIEWER COMMENTS

Reviewer #1 (Remarks to the Author):

In this computational study, Pan-Pan Chen et al. discuss the mechanism and selectivity-controlling factors of the Rh-catalyzed Annulative Cleavage of Bicyclo[1.1.0] butanes. The manuscript is well written and provides new and important insights into the selectivity of this transformation. Hence, it could be suitable for nature communications. However, I have a few remarks that the authors should address before the manuscript could be considered for publication.

Major points:

- 1) Based on the computed profiles, Rh(I)/dppe is significantly less active than Rh(I)/PPh₃ for the cycloisomerization of BCB (the reaction barriers for the corresponding rate determining steps are 30.7 and 22.8 for Rh(I)/dppe and Rh(I)/PPh₃, respectively). Is this consistent with the experimentally available kinetic data?
- 2) Thermal corrections were computed at 298K. However, a reaction barrier of ~31 kcal/mol is not realistic under these conditions.
- 3) The authors argue that their results will facilitate the design of new catalytic reactions. The manuscript would benefit from a more detailed discussion of this aspect. In relation to this point, the authors could elaborate on the relative importance of steric vs electronic effects on the selectivity of this transformation. Do the authors expect similar selectivities with different monodentate and bidentate ligands?

Minor points:

- 1) The authors should indicate the energy units in the captions of the computed energy profiles.
- 2) The chemical significance of 1-2% changes in the atomic orbitals coefficients could be questioned. The authors could corroborate their conclusions with additional interpretation tools.

Reviewer #2 (Remarks to the Author):

This Communication by Chen, Wipf and Houk describes a detailed computational mechanistic study of a divergent Rh(I)-catalyzed cycloisomerization of bicyclobutanes previously reported by the Wipf group in 2008 (ref 33), in order to explain the drastic ligand effect observed on the nature of the products formed (pyrrolidine vs azepane) and on the diastereoselectivity. Various mechanistic pathways involving either neutral or zwitterionic intermediates have been considered and computationally evaluated for their validity. In the original publication (ref 33), a divergent formation of two distinct metal-carbene species were proposed to be responsible for such an effect (similar to species D and H here), but confirmation via calculations was not performed and the question of ligand effect and diastereoselectivity remained. In this work, it was determined that oxidative addition and carbene formation occur in a concerted manner without the formation of a species such as C, and the nature of the ligand determines the outcome based on coordinative saturation of the Rh(I) center, where a ligand such as PPh₃ leaves a free coordination site for alkene coordination and direction of the site of metal-carbene formation, eventually leading to either a pyrrolidine (with PPh₃) or an azepane (with dppe). The quality of the work is very high, the manuscript is clearly written, and the impact of the conclusions drawn herein is significant to other transformations employing bicyclobutanes as strain-release reagents in transition metal catalysis. For these reasons, I would recommend that this work should be accepted in Nature Communications after the following minor corrections have been addressed:

1. The reason for product divergence depending of the ligand (e.g. coordinative saturation allowing the alkene to direct carbene formation in the case of PPh₃) should be more clearly stated in the conclusion as well as in page 5 (bottom right) when discussing the azepane pathway. From this reviewer's point of view, this is one of the most important conclusions and should be clear to maximize the impact of the work.
2. Page 1, column 1 (bottom): the expression Rh/PPh₃ and Rh/dppe should be replaced by [Rh(C₂H₄)₂Cl]₂/PPh₃ and [Rh(CO)₂Cl]₂/dppe to be more precise.
3. Page 1, column 2 (bottom): 'Oxidative addition' should be replaced by 'Endo oxidative addition', as exo pathways (S_N2-like) have also been considered with other metals such as Pd (see ref 30).
4. Page 2, Fig. 2: Curved arrows shown in red for carbene formation in paths A and D should be redrawn for improved clarity.
5. Page 2, column 1 (top): 'paths C and D' should be 'paths D and E'. Please verify and modify accordingly.

6. Page 4, column 2: 'Both TS14 and 15 are highly disfavored because the high-polarity zwitterionic species is disfavored in the experimentally employed low-polarity toluene solvent.'

This statement is too bold without supporting evidence provided and should be moderated by expressions such as 'likely' or 'presumably'.

The study by Wipf and Houk is an in-depth computational study on a significant methodology involving strained carbocycles as "spring-loaded" molecules. Exploration of the origin of selective access to internal versus external carbenoids based on choice of ligand on the rhodium catalyst using DFT provides valuable insight that will lead to exciting applications for this methodology. The reviewer recommends publication of this manuscript in *Nature Communications* after the authors have satisfactorily addressed two main concerns described below. In addition, some minor suggestions are also included that need to be considered by the authors.

Major Issues

1. Since the major claim of this study is that the ligand-influences the mechanistic pathway and product outcome, the major point that I believe must be addressed before accepting the manuscript is the calculation of each concerted carbenoid pathway (as shown in Fig 4 and Fig 7) with both ligands. The authors assume an L1 species for PPh₃ and L2 species (bidentate binding) for DPPE. This isn't necessarily true. This is especially significant if the lower energy TS5 is accessible for 'monoligated' DPPE (coordination of Rh to only one of the phosphorous atoms).
2. Has PBE0-D3 previously been used as a functional for single-point energy calculations in rhodium carbene systems? The reviewer would like to either see references that justify the use of this method or some benchmarking studies that led to the choice of method.

Other minor issues

3. The lettering in Fig 2 is very confusing, consequently the text describing these pathways are also difficult to follow. For example, page 2 paragraph 1, " Paths A and B converge on internal carbenoid (D) formation, and paths C and D converge on external carbenoid (H) formation." This is extremely confusing as a reader, especially since carbenoid D is not part of path D. There must be some type of distinction between paths and species, perhaps use numbers for the species and then letters for the paths?
4. Please write more detailed figure captions. For example, it would be extremely useful to indicate which path (from Fig 2 and Fig 3) each energy profile corresponds to (please add a description to the caption or directly into the figure).
5. In Fig 8a, it might be helpful to switch TS 18 to the left and TS18* to the right and reference TS18* +4.0 kcal/mol. The current format of Fig 8a doesn't follow the patterns in the rest of the manuscript (Fig 6a, 6b, 8b).
6. The SI had major issues. The CYL structures in many of the figures were cut off (see screenshot below). For the SI, it would also be nice if the subtitle # lined up with the figure #. It's a little bit confusing to determine which section/figure the manuscript is referring to.

2. DFT-optimized structures of selected intermediates and transition states involved in Rh(I)/PPh₃ catalysis

Supplementary Fig. 1 DFT-optimized structures of selected intermediates and transition states involved in Rh(I)/PPh₃-catalyzed cycloisomerizations. L = PPh₃.

October 7, 2022

Dear editor, dear reviewers,

Re. “**How Mono- and Diphosphine Ligands Alter Regioselectivity of the Rh-Catalyzed Annulative Cleavage of Bicyclo[1.1.0]butanes**”

Pan-Pan Chen¹, Peter Wipf^{2*}, and K. N. Houk^{1*}

¹Department of Chemistry and Biochemistry, University of California, Los Angeles, California 90095, United States. ²Department of Chemistry, University of Pittsburgh, 219 Parkman Avenue, Pittsburgh, Pennsylvania 15260, United States.

*E-mail: pwipf@pitt.edu; houk@chem.ucla.edu.

Thank you for the time you have spent on our manuscript (NCOMMS-22-29624-T). We are delighted with the comments of the three reviewers and we would like to thank them for their valuable advice and positive suggestions. I am pleased to say that we have done our best to address all the pertinent issues raised by the reviewers. We have highlighted in yellow the changes made to the manuscript and revised the Supplementary Information accordingly. Please find attached a revised manuscript for consideration for publication as an article in *Nature Communications* and, below, a point-by-point response to the reviewers' comments:

Reviewer #1 (Remarks to the Author):

In this computational study, Pan-Pan Chen et al. discuss the mechanism and selectivity-controlling factors of the Rh-catalyzed Annulative Cleavage of Bicyclo[1.1.0] butanes. The manuscript is well written and provides new and important insights into the selectivity of this transformation. Hence, it could be suitable for nature communications. However, I have a few remarks that the authors should address before the manuscript could be considered for publication.

Major points:

1) Based on the computed profiles, Rh(I)/dppe is significantly less active than Rh(I)/PPh₃ for the cycloisomerization of BCB (the reaction barriers for the corresponding rate determining steps are 30.7 and 22.8 for Rh(I)/dppe and Rh(I)/PPh₃, respectively). Is this consistent with the experimentally available kinetic data?

Response: Experimentally, when using Rh(I)/PPh₃ as catalyst, the optimal reaction yield for substrate **1** could be achieved within 15 minutes, while when employing Rh(I)/dppe as catalyst, the reaction time needed to be doubled to 30 minutes to obtain >80% conversion to product. It is possible that Method A (using PPh₃ as a ligand) proceeds even more quickly, and at lower temperatures, but this has not yet been experimentally tested.

Conditions:

Method A: $[\text{Rh}(=\text{O})_2\text{Cl}]_2$, Ph_3P , PhMe , 110°C , 15 min, 87%

Method B: $[\text{Rh}(\text{CO})_2\text{Cl}]_2$, dppe , PhMe , 110°C , 30 min, 83%

In a control reaction, the isomerization of a BCB (**1**) to a 1,3-diene (**4**), the same trend was observed, and here the reaction time difference was more pronounced:

Conditions:

Method A: $[\text{Rh}(=\text{O})_2\text{Cl}]_2$, Ph_3P , PhMe , 110°C , 15 min, 60%

Method B: $[\text{Rh}(\text{CO})_2\text{Cl}]_2$, dppe , PhMe , 110°C , 90 min, 58%

Using $\text{Rh}(\text{I})/\text{PPh}_3$ as catalyst, the ring opening and isomerization required 15 minutes or less, while with $\text{Rh}(\text{I})/\text{dppe}$ as catalyst, 90 minutes reaction time were necessary to obtain an equivalent conversion to product.

The large difference in reaction barrier that we found in the computational studies has motivated us to further investigate experimental conditions in our continuation of this work. Furthermore, it is also quite possible that the conversion of the pre-catalyst, $[\text{Rh}(=\text{O})_2\text{Cl}]_2$ to the active catalyst has a reaction barrier exceeding 22.8 kcal/mol.

2) Thermal corrections were computed at 298K. However, a reaction barrier of ~ 31 kcal/mol is not realistic under these conditions.

Response: Thanks for this comment. We generally report free energies (including thermal corrections) at a standard state of 1M and 298 K. Indeed, we agree with the reviewer that a reaction barrier of 31 kcal/mol is insurmountable at room temperature (298K), but it is overcome at 110°C , the actual experimental reaction temperature.

3) The authors argue that their results will facilitate the design of new catalytic reactions. The manuscript would benefit from a more detailed discussion of this aspect. In relation to this point, the authors could elaborate on the relative importance of steric vs electronic effects on the selectivity of this transformation. Do the authors expect similar selectivities with different monodentate and bidentate ligands?

Response: Thanks for this comment, and we do agree with the reviewer that selectivity is the result of both steric and electronic effects. The major claim of our study is that the ligand influences the mechanistic pathway and product outcome, which are related to ligand-controlled regioselectivity. For a monodentate ligand, an L1 species, coordinative saturation is enabling the alkene in the substrate acting as an auxiliary ligand to coordinate with the metal, promoting carbon-carbon bond cleavage and directing carbenoid formation. As a result, carbenoid formation is an intramolecular reaction, and internal carbenoid formation is favored; thus, for monodentate ligands, electronic effects have a greater influence on selectivity than steric effects. Based on this, different monodentate ligands showed similar selectivity (*J. Am. Chem. Soc.* **130**, 6924–6925 (2008), Table 1).

In the case of bidentate ligand, the catalyst coordination saturates, carbon-carbon bond cleavage coupled with carbenoid formation is an intermolecular reaction. Consequently, the catalyst tends to attack the external carbon, minimizing steric hindrance. Therefore, for bidentate ligands, steric effects contribute more to selectivity than electronic effects, leading to similar selectivity with different bidentate ligands (*J. Am. Chem. Soc.* **130**, 6924–6925 (2008), Table 1).

Finally, the realization of the ligand effects and the accompanying reaction barriers have inspired us to consider the use of different pre-catalysts (to decrease the influence of the conversion of pre-catalyst to active catalyst) and to study the product-directing effects of intramolecularly coordinating (auxiliary) ligands, in the hope to also identify new reaction pathways to novel rearrangement products.

We have added above discussions into the revised manuscript.

Minor points:

1) The authors should indicate the energy units in the captions of the computed energy profiles.

Response: Thanks for this comment. We have made modifications accordingly in the revised manuscript.

2) The chemical significance of 1-2% changes in the atomic orbitals coefficients could be questioned. The authors could corroborate their conclusions with additional interpretation tools.

Response: Thanks for this comment. We agree with the reviewer that 1-2% changes in the atomic orbitals coefficients are not sufficient to produce selectivity (regioselectivity), and in fact, in the manuscript, we did not intend to suggest such a small difference in atomic orbital coefficients to be responsible for regioselectivity. In the Rh/PPh₃ case, the key difference between the two competing transition states is the skeletal deformation. The internal carbenoid formation transition state is favored due to lower strain. In contrast, with Rh/dppe, the external carbenoid formation transition state is favored since the substituents can assume a staggered conformation with respect to the breaking of the central C–C bond of the bicyclobutane. In order to avoid confusion, we have eliminated the discussions regarding the orbital interaction and atomic orbital coefficients part in the revised manuscript.

Reviewer #2 (Remarks to the Author):

This Communication by Chen, Wipf and Houk describes a detailed computational mechanistic study of a divergent Rh(I)-catalyzed cycloisomerization of bicyclobutanes previously reported by the Wipf group in

2008 (ref 33), in order to explain the drastic ligand effect observed on the nature of the products formed (pyrrolidine vs azepane) and on the diastereoselectivity. Various mechanistic pathways involving either neutral or zwitterionic intermediates have been considered and computationally evaluated for their validity. In the original publication (ref 33), a divergent formation of two distinct metal-carbene species were proposed to be responsible for such an effect (similar to species D and H here), but confirmation via calculations was not performed and the question of ligand effect and diastereoselectivity remained. In this work, it was determined that oxidative addition and carbene formation occur in a concerted manner without the formation of a species such as C, and the nature of the ligand determines the outcome based on coordinative saturation of the Rh(I) center, where a ligand such as PPh₃ leaves a free coordination site for alkene coordination and direction of the site of metal-carbene formation, eventually leading to either a pyrrolidine (with PPh₃) or an azepane (with dppe). The quality of the work is very high, the manuscript is clearly written, and the impact of the conclusions drawn herein is significant to other transformations employing bicyclobutanes as strain-release reagents in transition metal catalysis. For these reasons, I would recommend that this work should be accepted in Nature Communications after the following minor corrections have been addressed:

1. The reason for product divergence depending of the ligand (e.g. coordinative saturation allowing the alkene to direct carbene formation in the case of PPh₃) should be more clearly stated in the conclusion as well as in page 5 (bottom right) when discussing the azepane pathway. From this reviewer's point of view, this is one of the most important conclusions and should be clear to maximize the impact of the work.

Response: Thanks for this comment. We have updated the corresponding discussions in the revised manuscript.

Newly added text when discussing the azepane pathway: The activation mode through **TS5** or **TS5***, involves the alkenyl group acting as auxiliary ligand bridging to the metal center. By contrast, the bidentate ligand dppe coordinatively saturates the Rh, and thus does not require the assistance of the alkene during the cleavage of the carbon-carbon bond.

Newly added text in conclusion: In the regioselectivity-determining C–C bond cleavage and Rh carbenoid formation step with the PPh₃ ligand, the alkene auxiliary ligand coordinates with the metal, facilitating carbon-carbon bond cleavage, and guiding carbenoid formation. The dppe ligand saturates the metal coordination site, and the alkene in the substrate is not necessary as assisting ligand. Accordingly, different factors control the regioselectivity, and the reaction paths.

2. Page 1, column 1 (bottom): the expression Rh/PPh₃ and Rh/dppe should be replaced by [Rh(C₂H₄)₂Cl]₂/PPh₃ and [Rh(CO)₂Cl]₂/dppe to be more precise.

Response: Thanks for this comment. We have made modifications in the revised manuscript.

3. Page 1, column 2 (bottom): 'Oxidative addition' should be replaced by 'Endo oxidative addition', as exo pathways (S_N2-like) have also been considered with other metals such as Pd (see ref 30).

Response: Thanks for this comment. We have made modifications in the revised manuscript.

4. Page 2, Fig. 2: Curved arrows shown in red for carbene formation in paths A and D should be redrawn for improved clarity.

Response: Thanks for this comment. We have made modifications in the revised manuscript.

5. Page 2, column 1 (top): 'paths C and D' should be 'paths D and E'. Please verify and modify accordingly.

Response: Thanks for this comment, and the reviewer is right. We have made modifications accordingly in the revised manuscript.

6. Page 4, column 2: 'Both TS14 and 15 are highly disfavored because the high-polarity zwitterionic species is disfavored in the experimentally employed low-polarity toluene solvent.' This statement is too bold without supporting evidence provided and should be moderated by expressions such as 'likely' or 'presumably'.

Response: Thanks for this comment. We have made modifications accordingly in the revised manuscript.

Reviewer #3 (Remarks to the Author):

The study by Wipf and Houk is an in-depth computational study on a significant methodology involving strained carbocycles as "spring-loaded" molecules. Exploration of the origin of selective access to internal versus external carbenoids based on choice of ligand on the rhodium catalyst using DFT provides valuable insight that will lead to exciting applications for this methodology. The reviewer recommends publication of this manuscript in Nature Communications after the authors have satisfactorily addressed two main concerns described below. In addition, some minor suggestions are also included that need to be considered by the authors.

Major Issues

1. Since the major claim of this study is that the ligand-influences the mechanistic pathway and product outcome, the major point that I believe must be addressed before accepting the manuscript is the calculation of each concerted carbenoid pathway (as shown in Fig 4 and Fig 7) with both ligands. The authors assume an L1 species for PPh₃ and L2 species (bidentate binding) for DPPE. This isn't necessarily true. This is especially significant if the lower energy TS5 is accessible for 'monoligated' DPPE (coordination of Rh to only one of the phosphorous atoms).

Response: Thanks for this comment. For the Rh/PPh₃ case, in the main reactions of the experiment, the ratio of catalyst to ligand is 1 to 1, thus, a metal center can have only one monodentate phosphine ligand coordination. In the regioselectivity-determining step, coordinative saturation is enabling the alkene in the substrate acting as an auxiliary ligand to coordinate with the metal, facilitating carbon-carbon bond cleavage, and directing the carbenoid formation. Based on this scenario, the competition between **TS5** and **TS5*** (Fig. 1) determines the regioselectivity, consistent with our conclusions shown in the manuscript (Fig. 6a in the revised manuscript).

Fig. 1. Alternative transition states for rhodium carbenoid formation involved in Rh/PPh₃ case.

For the Rh/dppe case, in addition to **TS18** and **TS18***, we also calculated the alternative transition states **TS18a** and **TS18b**, corresponding to external and internal carbenoid formation transition state, respectively. As shown in Fig. 2, for the external carbenoid formation, **TS18a** is 15.5 kcal/mol higher in energy than **TS18**. Similarly, for internal carbenoid formation, **TS18b** is 7.4 kcal/mol higher in energy than **TS18***. These results indicate that the competition between **TS18** and **TS18*** determines the regioselectivity, which is consistent with our conclusions shown in the manuscript (Fig. 8a in the revised manuscript). The higher energies of **TS18a** and **TS18b** are due to the fact that in order to accommodate alkene coordination to the rhodium, the bidentate ligand dissociates one phosphine coordination, resulting in a large catalyst distortion, eventually leading to a disfavored transition state. We have included this part of discussions into the revised Supplementary Information (Supplementary Fig. 12 in the revised Supplementary Information).

Fig. 2. Alternative transition states for rhodium carbenoid formation involved in Rh/dppe case.

2. Has PBE0-D3 previously been used as a functional for single-point energy calculations in rhodium carbene systems? The reviewer would like to either see references that justify the use of this method or some benchmarking studies that led to the choice of method.

Response: Thanks for this comment. In organometallic reaction systems, PBE0 is one of the reliable density functionals for the energy calculations and shows a good performance with the addition of Grimme's D3 dispersion corrections. For selected references, please see: (1) Dohm, S., Hansen, A., Steinmetz, M., Grimme, S. & Checinski, M. P. Comprehensive Thermochemical Benchmark Set of Realistic Closed-Shell Metal Organic Reactions. *J. Chem. Theory Comput.* **14**, 2596–2608 (2018). In this paper, reactions containing rhodium have been chosen as part of benchmark set. (2) Husch, T., Freitag, L. & Reiher, M. Calculation of Ligand Dissociation Energies in Large Transition-Metal Complexes. *J. Chem. Theory Comput.* **14**, 2456–2468 (2018). (3) Steinmetz, M. & Grimme, S. Benchmark Study of the Performance of Density Functional Theory for Bond Activations with (Ni,Pd)-Based Transition-Metal Catalysts. *ChemistryOpen.* **2**, 115–124 (2013).

In our study, we performed calculations regarding regioselectivity with different single-point methods. As shown in the Table 1, all the tested methods showed consistent selectivity and were consistent with the experimental observations. Therefore, based on our calculations and literature reports, we selected PBE0-D3 as the single-point calculation method. We have included this part of discussions into the revised Supplementary Information (Supplementary Table 1 in the revised Supplementary Information).

Table 1. Methods benchmark for energies. Geometries of the structures are optimized with B3LYP-D3 functional. Energies are in kcal/mol.

Other minor issues

3. The lettering in Fig 2 is very confusing, consequently the text describing these pathways are also difficult to follow. For example, page 2 paragraph 1, " Paths A and B converge on internal carbenoid (D) formation, and paths C and D converge on external carbenoid (H) formation." This is extremely confusing as a reader, especially since carbenoid D is not part of path D. There must be some type of distinction between paths and species, perhaps use numbers for the species and then letters for the paths?

Response: Thanks for this comment. We have made modifications accordingly in the revised manuscript. In the updated manuscript, we use Roman numerals for species and then letters for paths (Fig.2 and Fig. 3 in the revised manuscript).

4. Please write more detailed figure captions. For example, it would be extremely useful to indicate which path (from Fig 2 and Fig 3) each energy profile corresponds to (please add a description to the caption or directly into the figure).

Response: Thanks for this comment. We have made modifications accordingly in the revised manuscript.

5. In Fig 8a, it might be helpful to switch TS 18 to the left and TS18* to the right and reference TS18* +4.0 kcal/mol. The current format of Fig 8a doesn't follow the patterns in the rest of the manuscript (Fig 6a, 6b, 8b).

Response: Thanks for this comment. We have made modifications accordingly in the revised manuscript.

6. The SI had major issues. The CYL structures in many of the figures were cut off (see screenshot below). For the SI, it would also be nice if the subtitle # lined up with the figure #. It's a little bit confusing to determine which section/figure the manuscript is referring to.

Response: Thanks for this comment. We have made modifications accordingly in the revised Supplementary Information.

In closing, I hope that you find these explanations, comments and changes address many of reviewer's concerns. We believe that the review process has improved this paper. We are hopeful that you will find the revised version suitable for publication. If there are any outstanding concerns or points you would wish us to address in more detail, please do not hesitate to let me know.

All the best,

Ken

REVIEWERS' COMMENTS

Reviewer #1 (Remarks to the Author):

The authors properly addressed all comments - I believe the manuscript can be accepted in its current form.

Reviewer #2 (Remarks to the Author):

This is a re-submission of a computational study regarding the divergent Rh(I)-catalyzed cycloisomerization of bicyclobutanes. Previous concerns raised by this reviewer along with minor corrections have all been addressed, so the manuscript is ready for acceptance in Nature Communications in its current form.

Reviewer #3 (Remarks to the Author):

The authors have satisfactorily addressed the reviewers' comments. The additions to the manuscript have improved the clarity of the study and highlighted the impact of the work. I highly recommend the manuscript for publication in Nature Communications.